# Engineering the Metabolic Landscape of Microorganisms for Lignocellulosic Conversion

**DOI:** 10.3390/microorganisms11092197

**Published:** 2023-08-31

**Authors:** Julián Mario Peña-Castro, Karla M. Muñoz-Páez, Paula N. Robledo-Narvaez, Edgar Vázquez-Núñez

**Affiliations:** 1Centro de Investigaciones Científicas, Instituto de Biotecnología, Universidad del Papaloapan, Tuxtepec 68301, Oaxaca, Mexico; julianpc@unpa.edu.mx; 2CONAHCYT—Instituto de Ingeniería, Unidad Académica Juriquilla, Universidad Nacional Autónoma de México, Queretaro 76230, Queretaro, Mexico; kmunozp@iingen.com.mx; 3Tecnológico Nacional de Mexico/ITS de Tierra Blanca, Tierra Blanca 95180, Veracruz, Mexico; p.robledo@itstb.edu.mx; 4Grupo de Investigación Sobre Aplicaciones Nano y Bio Tecnológicas para la Sostenibilidad (NanoBioTS), Departamento de Ingenierías Química, Electrónica y Biomédica, División de Ciencias e Ingenierías, Universidad de Guanajuato, Lomas del Bosque 103, Lomas del Campestre, León 37150, Guanajuato, Mexico

**Keywords:** bioproducts, metabolic engineering, lignocellulosic residues, sustainability

## Abstract

Bacteria and yeast are being intensively used to produce biofuels and high-added-value products by using plant biomass derivatives as substrates. The number of microorganisms available for industrial processes is increasing thanks to biotechnological improvements to enhance their productivity and yield through microbial metabolic engineering and laboratory evolution. This is allowing the traditional industrial processes for biofuel production, which included multiple steps, to be improved through the consolidation of single-step processes, reducing the time of the global process, and increasing the yield and operational conditions in terms of the desired products. Engineered microorganisms are now capable of using feedstocks that they were unable to process before their modification, opening broader possibilities for establishing new markets in places where biomass is available. This review discusses metabolic engineering approaches that have been used to improve the microbial processing of biomass to convert the plant feedstock into fuels. Metabolically engineered microorganisms (MEMs) such as bacteria, yeasts, and microalgae are described, highlighting their performance and the biotechnological tools that were used to modify them. Finally, some examples of patents related to the MEMs are mentioned in order to contextualize their current industrial use.

## 1. Introduction

Society faces a growing demand for sustainable and renewable energy sources, which are required to meet the growing energy needs of the global population. The use of starch and lignocellulosic residues, such as agricultural waste, forestry residues, and industrial by-products, as a source of biofuels and high-value chemicals has been attracting research efforts in recent years [1,2].

Lignocellulosic residues are a rich source of polymeric carbohydrates such as cellulose, hemicellulose, pectin, and suberin. Another abundant component is lignin, a complex polyaromatic molecule synthesized from amino acids. Microbes have the ability to metabolize both classes of polymers into products [3]. However, lignocellulose evolved to be recalcitrant to microbial attacks and provide protection to plants against pathogens. Therefore, the use of chemical and biological processes is being researched to overcome this natural barrier.

Metabolic engineering (ME) is the process of modifying the biochemical pathways of living cells to yield a desired product and is an attractive tool to optimize waste-to-product conversion. ME is a field of genetic engineering (GE) that was pioneered by the introduction of new genes into cells to alter their metabolism or by the mutation or replacement of existing genes within cells, both by recombinant and traditional mutagenesis techniques. The use of metabolic engineering has been widely applied to various microorganisms, including bacteria, yeast, and microalgae, for the production of biofuels and high-value chemicals [4], as well as in other areas of application such as brewing [5] and remediation [6].

The primary goal of metabolic engineering in lignocellulosic conversion is to optimize the metabolic pathways of microorganisms to increase the yield of biofuels and high-value chemicals [7]. In recent years, more complex traits such as the removal of substrate inhibition, toxicity avoidance, and improvements to auxiliary pathways have been added to these objectives.

With the advent of economic DNA-sequencing methodologies, metagenomics has helped us realize the diversity and complexity of the metabolic landscape of microorganisms; such networks of metabolic pathways determine the ability of the cells to produce specific products [8]. To achieve the optimal conversion of lignocellulosic residues into high-value chemicals, it is essential to have a deep understanding of the metabolic capacities of the microorganisms used for the conversion process. This knowledge is critical for the effective design and implementation of metabolic engineering strategies to enhance conversion efficiency and yield.

In recent years, significant advances have been made in the field of metabolic engineering, especially the introduction of the concept of laboratory evolution, either of complete organisms or selected enzymes [9]. This is leading to the development of new and improved metabolic pathways for the conversion of lignocellulosic residues and CO_2_ management. Despite these innovations, there are still significant challenges that need to be overcome to further enhance the efficiency and yield of the conversion process. These challenges include the development of improved metabolic pathways, the identification of novel enzymes and metabolic pathways, and the optimization of the metabolic landscape of the microorganisms used for the conversion process.

Metabolic engineering has the potential to play a significant role in the development of sustainable and renewable energy sources. The metabolic landscape of microorganisms is a critical component in the conversion of lignocellulosic residues into biofuels and high-value chemicals, and further advances in metabolic engineering are essential to enhance the efficiency and yield of the conversion process. The goal of this review is to look at where traditional and cutting-edge metabolic engineering for lignocellulosic conversion is now, what new methods are being used, and what the future looks like for this field. This will show how this field could help develop sustainable and renewable energy sources.

## 2. Traditional Techniques for Converting Lignocellulosic Residues into High-Value Products

Currently, the preferred methods for biomass pretreatments are chemical conversions such as acid or basic hydrolysis, steam explosion, hydrothermolysis, ion liquids, and organosolv [10]. They allow processing in high volumes, the rapid deconstruction of the cell wall structure, and the controlled release of substrates.

Hydrolisis, either basic or acid, which involves the use of chemicals such as acids, alkalis, organic solvents, and ionic liquids, has been reported to significantly impact the structure of lignocellulosic biomass [11]. The composition of the biomass determines the effectiveness of the alkaline solutions; among the most common are NaOH, KOH, and Ca(OH)_2_, but lately the use of hydrazine and anhydrous ammonia has been proven to be successful. It has been attributed to the swelling of the biomass, the increase in its internal surface area, and the decrease in the degree of polymerization and crystallinity of cellulose [12]. Alkaline pretreatment breaks the linkage between lignin and other carbohydrate fractions in lignocellulosic biomass, making the carbohydrates in the heteromatrix more accessible. Regarding acidic solutions, sulfuric, phosphoric, and hydrochloric acids have been demonstrated to successfully hydrolize biomass at dilute concentrations. One important drawback of using acidic solutions is their high corrosivity, which represents a limitation in technical and economic terms [13].

Organosolv is a biomass pretreatment in which organic or aqueous-organic solvents are used. The application of this pretreatment requires the use of catalysts such as HCl or H_2_SO_4_. Among the most common organic acids used as catalyzers are oxalic, acetylsalicylic, and salicylic acids; these catalyzers enhance the solubilization of hemicellulose and lignin, facilitating access to the exposed cellulose. The reaction occurs at high temperatures ranging from 100 to 250 °C, and the use of solvents with a low melting point is preferred. The main advantages of using organosolv pretreatment are its high selectivity, ease of processing, and obtaining fractions of dry lignin, aqueous hemicellulose, and pure cellulose that can be used in downstream conversions [14].

Physical-based treatments, such as steam explosion (SE), are the most commonly applied methods of biomass pretreatment. SE involves physically preconditioned biomass being treated with high-pressure saturated steam at temperatures between 160 and 240 °C and pressures between 0.7 and 4.8 MPa. The physical effect of vapor makes fibers “explode”, hydrolyzing them to obtain hemicellulose in the liquid phase. The high temperature transforms lignin, making the cellulose in the solid fraction more accessible and increasing the digestibility of the lignocellulosic feedstock [15].

In a similar way to steam explosion, liquid hot water (LHW) or thermolysis pretreatment employs water at an elevated temperature. Similar results to SE regarding the composition of the fractions are obtained by applying LHW [16].

A new physicochemical approach has been developed called ionic liquid (IL). This pretreatment uses low melting point solvents with high polarities and negligible vapor pressures. Most ILs used in biomass fractionation are imidazonium salts. Studies have shown that 1-allyl-3-methylimidazonium chloride (AMIMCl) and 1-butyl-3-methylimidazonium chloride (BMIMCl) are effective as non-derivatizing solvents for dissolving cellulose at temperatures below 100 °C [17].

The deconstruction of biomass and the subsequent production of biofuels and by-products can be achieved using separate hydrolysis and fermentation (SHF) by applying improved enzymatic pretreatments [4] on wild-type or genetically improved plant biomass [2]. Both options work to deconstruct the complex biomass for the release of monomers and metabolites that can be used by other organisms for subsequent bioconversion. However, simultaneous saccharification and fermentation (SSF) and consolidated bioprocessing (CBP) can be achieved with improved microorganisms and enzymes with tolerance to diverse stresses such as acidic pH, toxic substrates, or thermotolerance [18]. Both native, evolved, and genetically modified microorganisms (GMOs) and their combinations can be utilized in these methods, along with their corresponding enzymatic processes, to achieve high yields of bioproducts.

Improvements to these technological approaches, such as increasing yield and productivity, reducing enzymatic inhibition, and expanding the availability of microorganisms capable of bioprocessing under different conditions, are being actively researched. The recent development of technologies designed to access and manipulate genetic information has opened up new avenues for overcoming limitations in the bioconversion of biomass into biofuels and bioproducts. In order to better appreciate the power of mass DNA and RNA sequencing and evolution techniques, we need to review historically relevant breakthroughs in traditional ME.

## 3. Enhancing Biomass-Based Products through Metabolic Engineering

Living biological systems are fundamentally made up of information molecules and a series of enzymatic reactions. Numerous interconnected metabolic pathways simultaneously synthesize and break down thousands of organic macromolecules during cellular metabolic activity. In consequence, the expression of a native catabolic or anabolic pathway can be increased to boost the production of target compounds in an organism, or a pathway can be transferred from another organism [19]. Additionally, genes can be knocked out using techniques such as homologous recombination [20] or RNAi, which aims to reduce the mRNA of the unwanted protein [21]. This can reduce carbon leakage and stop the production of unwanted compounds.

To increase the yield of desired products, two approaches can be taken. The first is to change the expression of regulatory genes that control the protein-coding genes downstream of multiple biosynthesis genes. The second is to change the expression of enzyme-coding genes to navigate limiting steps in the target pathway, slow down the breakdown of target compounds, and stop competing pathways [22].

By utilizing ME, the biosynthesis of biomass-based products can be greatly enhanced. Traditional methods include direct selection from nature [23], random physicochemical mutagenesis [24], and screening to find microbial strains that could increase bioprocessing capabilities [25]. However, there can be limitations such as unpredictability, slowness, and a lack of control. With the application of ME techniques, the performance of microbes can be improved, leading to the development of microorganisms with enhanced biomass conversion abilities and overcoming the limitations of traditional methods.

Using genetic engineering (GE) techniques can enhance the enzymatic machinery and increase the conversion of biopolymers into monomers for fermentation. ME can also expand the ability to utilize a wider range of carbon sources and increase resistance to inhibitors and harsh environmental conditions. By manipulating the production and regulation of specific enzymes and eliminating counterproductive pathways, specific biochemical routes can be improved [26,27]. ME can also introduce a new metabolic pathway into a host organism, with the advantage of modifying the microbial species without causing an accumulation of undesirable mutations [28].

Bacteria, yeast, and microalgae have been extensively studied for their potential to convert renewable biomass into high-added-value products. The metabolic pathways of these microorganisms have been manipulated to optimize the production of specific by-products. The use of these microorganisms as biofactories is attractive due to their rapid growth rate, high metabolic flexibility, and ability to produce a wide range of products. Their products have the potential to replace petroleum-based materials with sustainable, renewable alternatives and contribute to a greener and more sustainable future.

### 3.1. Bacteria

Bacteria play a crucial role in the production of biofuels, as they are a highly genetically diverse group of microorganisms capable of converting various substrates into fuels such as bioethanol, biobutanol, biodiesel, and others. The workhorses of GMO bacteria include species such as *Zymomonas mobilis*, *Escherichia coli*, and *Bacillus subtilis*.

*Zymomonas mobilis* is a well-known producer of bioethanol [21], but it has the drawback of not being able to use pentose sugars, which are abundant in lignocellulosic feedstocks. To address this, Wang et al. [29] modified *Z. mobilis* to allow pentose utilization. A recombinant strain of *Z. mobilis* was developed that produced ethanol concentrations of 136 g/L in a solution of 295 g/L glucose without the need for any additions of amino acids or vitamins. The enhanced strain harbored multiple gene modules, i.e., the xylA/xlyB/tktA/talB operon for xylose utilization, the metB/yfdZ operon for lysine and methionine biosynthesis, and the thioesterase gene tesA to enhance free fatty acid biosynthesis for increased ethanol tolerance. This strain is now commercially available under the DuPont brand and is used for several biotechnological developments [27].

*E. coli* and *B. subtilis* have been chosen as biological models for ethanol production due to their ability to utilize diverse substrates. *E. coli* was one of the first microorganisms to be successfully modified through metabolic engineering [30]. Ajit et al. [31] transformed *E. coli* into a strain capable of producing bioethanol by adding genes from *Z. mobilis* encoding pyruvate decarboxylase and alcohol dehydrogenase. Further modifications led to the creation of a homoethanoleogenic derivative called the KO11 PPAL strain, which had improved expressions of heterologous pyruvate decarboxylase and alcohol dehydrogenase from *Z. mobilis*.

An example of the metabolic diversion of carbon is the strain of *B. subtilis* that was modified for bioethanol production through the disruption of the native lactate dehydrogenase gene and the addition of the *Z. mobilis* pyruvate decarboxylase and alcohol dehydrogenase II genes. This led to increased rates of cell growth and glucose consumption compared to the wildtype [32].

### 3.2. Yeasts

*Saccharomyces cerevisiae* is a microorganism with probably one of the longest histories of domestication and usage by humanity. With a genetic background that has been widely explored, manipulated, and tested at industrial scales, it is among the most popular biological platforms for producing bioethanol. Its metabolic routes have been modified using diverse genetic engineering tools, with a focus on gene regulatory systems, increasing stress tolerance, and improving transport mechanisms [33]. Although yeasts cannot directly convert some sugars (e.g., xylose) into alcohols, by using different metabolic alternatives, it has been possible to make yeast capable of co-fermenting non-native carbon sources. However, there are still limitations to overcome, such as low xylose fermentation yields compared to glucose and inhibitions due to toxic compounds in the hydrolysates.

For the production of biodiesel, oleaginous yeasts such as *Yarrowia lipolytica* can accumulate high levels of lipids and consume a wide range of lignocellulosic feedstocks while tolerating high inhibitor concentrations and operational conditions [34]. Qiao et al. [35] genetically modified strains of *Y. lipolytica* to produce high amounts of fatty acid methyl ester (FAME). They achieved this by creating synthetic pathways that helped funnel glycolytic NADH into NADPH and carbon into acetyl-CoA. The best strain achieved a productivity improvement of 25% over other engineered yeast strains. This improvement was due to the reduction in oxygen requirements caused by decreased NADH oxidation by aerobic respiration. Targeting the lipase-dependent pathway to the subcellular lipid body compartment resulted in a 10-fold higher fatty acid methyl ester (FAME) titer compared to the cytosolic pathway, and a combined simultaneous targeting of lipase pathways to the lipid body, endoplasmic reticulum, and peroxisome gave rise to the highest FAME titer of 1644.8 mg/L [36]. These strategies aimed at targeting metabolic pathways towards lipid bodies are expected to promote subcellular compartment engineering and offer new options for the biosynthesis of other lipid products.

One challenge in microbial oil production is the high production and extraction costs. Ledesma-Amaro et al. [37] proposed the possibility of secreting lipids into the culture broth to uncouple production and biomass processing. Two synthetic approaches were considered, one based on increasing fatty acids produced by enhancing flux through neutral lipid formation through the overexpression of diacylglycerol acyltransferase (DGA2) under the control of the pTEF promoter in the free fatty acids (FFAs) accumulating mutant Δ*faa1*Δ*mfe1* (disrupting acyl-CoA and B-oxidation). Simultaneously, *TGLA* (the intracellular lipase of *Y. lipolytica*) was overexpressed, resulting in a significant increase in FFA. The other strategy mimicked the bacterial system by a complete depletion of neutral lipid formation and by the expression of endogenous, re-localized heterologous acyl-CoA thioesterases in the same mutant (Δ*faa1*Δ*mfe1*) to produce free fatty acids. The results showed an increased lipid excretion of up to 85%. Remarkably, the engineered strains secreted a titer of 120% of dry cell weight in coupled fermentation.

### 3.3. Microalgae

Microalgae are a diverse and unique group of photosynthetic microorganisms that have the capability to produce a wide range of high-value commercial products, such as vitamins, antioxidants, omega-3 fatty acids, and immunostimulants [38]. Microalgae feature rapid proliferation and photoautrophic efficiency, converting CO_2_ and light into biomass and high-energy molecules such as lipids for biodiesel. Moreover, microalgae do not compete with food crops or agricultural land [39].

The production of biofuels from microalgae is a promising source of sustainable and economically viable environmental energy with low carbon emissions and the potential to replace limited fossil fuels. The composition of cells in microalgal cultures is influenced by environmental factors such as light, temperature, nutrient availability, pH, and salinity. In addition to the direct modification of metabolic pathways, genetic engineering involves the manipulation of enzymatic processes, transport, and photosynthetic functions to improve activities [40].

The transfer of the *Arabidopsis thaliana* WRINKLED 1 transcription factor to *Nannochloropsis salina* is an example of recombinant ME. This increased the lipid content by 36.5% and the FAME yield by 64%, making it a good choice for making biofuels in industrial microalgae [41]. Another case is the transformation of *Chlorella* sp. with the gene hydrogenase (HydA) mutated to avoid O_2_ enzymatic inhibition and increase H_2_ production. HydA was designed to prevent O_2_ from reaching the active site by substituting amino acid residues A105I, V256W, G113I, or V273I close to the gas tunnel, resulting in 7 to 30 times (depending on the tested O_2_ concentrations) increases in H_2_ production [42].

The targets for ME can be achieved through dedicated bioinformatic annotation, in particular with understudied organisms such as algae. In an interesting study, Sahoo et al. [43] examined the genomes of 26 species of microalgae belonging to various phylogenetic lineages and determined the complete sequence and structure of hundreds of hypothetical proteins coding for enzymes involved in lipid metabolism. This annotation included the development of 3D models with a special emphasis on active-site descriptions employing dynamic simulation methods. These findings have the potential to increase the oil content in microalgae, which is used for biofuel production, primarily biodiesel. Studies demonstrating the potential that microalgae have as a sustainable source of high-value commercial products and biofuels are presented in Table 1. Relevant examples include the organisms utilized, the engineering techniques applied, and the resulting improvements in bioproducts and yield. The successful cases of ME show the promise of further enhancing the metabolic landscape of microalgae to convert lignocellulosic residues into high-value products.

Most of the microbial capacities of interest for biomass conversion have their roots in natural metabolic activities that have been detected in ecological niches. Recently, with the advent of economical sequencing services, modern RNA/DNA-seq methodologies and derived databases can be incorporated into screenings [69]. This would help to overcome the historical bias for microorganisms that can be cultured in laboratory conditions and to incorporate into the biotechnological toolbox those uncultured genotypes that also have mechanisms of practical potential. Enzyme-coding genes, operons, and theoretical microbial genomes can be obtained using the large genomic databases currently available. For example, Naas et al. [70] used the well-studied system of cellulolytic organisms, mammal rumen combined with switchgrass adherence, and applied a metagenomic approach. In this way, the sequence of the genetic determinant is more important than the cellular context from which it originates. Metagenomes were assembled, and with the guidance of experimentally determined proteomes, the pathways of these non-culturable microorganisms were proposed. Extracellular carbohydrate-active enzymes (CAZymes), membrane transporters, multicellular membrane complexes, and biochemical pathways for the catabolic use of all carbohydrate-based components of plant cell walls (cellulose, xylose polysacharides, and mannans) were obtained. Finally, the use of these bioinformatically assembled sequences was illustrated with the recombinant expression (in this case from synthetic genes) of the most expressed cellulase and the characterization of its enzymatic activity, showing the feasibility of recovering functional activities from metagenomic studies and their combination with structural overlap with available registers in the Protein Data Bank.

The diversity of environments where microbes face evolutionary pressure to express active cell wall-degrading proteomes includes compost from agricultural wastes. Composting plant biomass is a common agricultural practice sustained by plant biomass decomposition. These fermentation techniques are ecological niches enriched with cellulolytic enzymes. Meneses et al. [71] built genomic libraries from red rice compost and screened them to find a cellulase. This bioinformatic analysis allowed the discovery of an open reading frame that was cloned and expressed as a recombinant to characterize it. As expected, when this cellulose was tested as an additive against the substrate where it was isolated, it released sugars that were later used as substrates by *S. cerevisae* to produce ethanol. Although the cell from which the cellulose was obtained is unknown, the logic of using it to lyse the original substrate proved a useful methodology.

Hyperthermophilic cellulolytic bacteria are an interesting subject of study and testing among the diversity of microorganisms that are suitable for the valorization of biomass. For example, *Caldicellulosiruptor bescii* is capable of producing exo- and endoglucanases, creating an exproteome that sustains growth in complex plant extracts with ethanol production. However, since its genome does not code for extracellular xylosidases, Kim et al. [72] complemented the exoproteome of *C. bescii* with xylose processing enzymes (xylosidase, endoglucanase, and xylanase) from other thermophilic bacteria and augmented growth and sugar release from model and plant-derived mixtures. The biological deconstruction of biomass is usually carried out with commercial enzyme cocktails. However, engineering the expression of these enzymes from the genome of suitable microorganisms (e.g., thermophiles) would decrease the cost of processing. This recombinant strategy helps to avoid adding costly external enzymes to the fermentation and allows the integral deconstruction of plant biomass into ethanol.

The use of thermostable enzymes in these processes allows for their direct use without the need for costly heat-transfer operations. Takeda et al. [73] applied experimental metagenomic techniques to samples obtained from hot spring sediments. The authors were able to construct sequences and determine the phylogenetics of digitally isolated glycoside hydrolase family members. The use of available protein crystallographic coordinates allowed the prediction of their structures and thermodynamic mechanisms of thermotolerance, as well as their physical production and enzymatic characterization through recombinant techniques.

In the future, the metabolic landscape of genetically modified organisms used in biomass conversion can be expanded by characterizing the molecular mechanisms governing the complex biological relations of natural niches. Kreuzenbeck et al. [74] approached the fungi–termites symbiotic relationship with biochemical, genetic, and analytical tools to analyze four different cellulolytic niches with a high diversity of organic molecules. The authors discovered dozens of chemical moieties related to terpenes of a diverse nature that were products of a rich enzymatic cocktail that could be partially reconstructed and included several cyclases. Characterizing the most frequently expressed sequence, which codes for a terpene cyclase, made it possible to suggest several conversion pathways that lead to terpenes of industrial interest, such as limonene or bisabolene. In addition to being interesting from a biological point of view, the relationship between fungi and insects could be used in the future to make fine chemicals from simple carbon substrates. This enzyme, which yielded multiple products, is a candidate for directed evolution to suppress or enhance substrate preferences or increase the yield of selected compounds.

With the advent of mass sequencing technologies, it is now possible to remotely screen ecological niches. Conspicuous niches to look after are hot springs. Different research groups have taken this prospective approach, and the raw sequences have been deposited in different databases [69]. Reichart et al. [75] bioinformatically explored the microbial diversity of hot springs for the discovery of extremozymes, both in pH and temperature. The authors found representatives of almost every CAZyme family in the dozens of samples screened. All the discovered CAZymes can be physically retrieved by standard PCR reactions or by artificial gene synthesis, a commercial methodology that has seen costs abate recently. A consequence of these types of research is the urgent global conservation of these unique ecosystems, which act as a living library of catalytic activities. Although celluloytic environments showed more diversity than hot springs, the CAZymes found in the latter are more likely to be thermoactive enzymes, rendering several candidates for biomass conversion catalysts. The source of novel CAZymes can also be non-traditional niches, for example, microbial eukaryotes. Chang and Lai [76] bioinformatically explored the family of glycoside hydrolases that encompass many members of biomass depolymerizing enzymes. These CAZymes present in metazoan genomes had representatives from every family of glycoside hydrolases. However, as the metazoan lineage advances toward multicellularity, these representatives decrease. Besides describing the evolutionary stages at which the loss of members of this family occurred and possible horizontal gene transfer events, they were also able to identify crustaceans as a potential source of such enzymes. The potential of these virtual isolates would be in conditions similar to the habitat of these organisms, such as high salinity and tolerance to denaturation.

Computer phylogenetic comparisons are the basis of decades-long research on enzyme and pathway discovery in microorganisms. The discovery of the initial sequence and its characterization are extended to new organisms by nucleotide or amino acid similarities. This knowledge has led to a new application of next-generation sequencing, which allows researchers to reconstruct genomes in a historically unprecedented manner and provides new contexts for discovering new enzymes. Piao et al. [77] highlighted the importance of bacterial genome organization in regulons. They also thought that uncharacterized genes next to genes that were already known to be CAZymes might also work on carbohydrates. The authors observed that a quarter of bacterial open reading frames still lack functional annotations and labeled this uncharacterized molecular information as “genomic dark matter”. After setting up a bioinformatic pipeline to digitally isolate cellulase-like candidates, dozens of predicted CAZymes were expressed in *E. coli* and tested on *Miscanthus* grass biomass to confirm their biochemical activity as sugar-releasing enzymes. This interesting research design allows for obtaining insight into nominal biochemical activity, investigating if they are part of operons, as well as their possible roles in their ecological niche. Some of these undiscovered sequences showed little or no activity in the tested conditions; however, interesting candidates for further improvement were obtained. These enzymes with low phylogenetic identity represent previously uncharacterized groups that may allow for the discovery of novel structures and mechanisms.

Sugar-based biomass has received the most attention in biological conversions to chemical products. However, as part of the plant cell wall, lignin is an abundant polymer built from aromatic rings derived from amino acids that can also provide carbon and electrons for cell metabolism. In nature, there are microbes with the genetic capacity to catabolize this complex polymer, which can be enriched and isolated as strains or as metagenomes in both prokaryotes and eukaryotes. That is the case in tropical forests, where the release of tons of lignified matter is continuously mobilized. Therefore, they must possess a microbial biome with the enzymatic ability to perform this catalysis. DeAngelis et al. [78] explored, through metagenomic techniques. the diversity of this niche and found that the transition of populations depends on the lignin load of the samples. Although the trend showed a loss of diversity in lignin-amended samples, the specialized metagenomes can be used in the future to retrieve the specific sequences of the measured enriched activities (e.g., peroxidases, oxidases, and cellobiohydrolases).

Once there is experimental evidence providing sufficient contextual information about a microorganism’s value as a testing chassis, its advancement is enriched by a genomic project. Through recombination or other mutagenesis methods, it is possible to improve strains capable of using lignin derivates, which can help make integral use of plant biomass. Morya et al. [79] isolated microbes from forest soil by challenging them to having several lignin-derivates such as vanillic, galic, and benzoic acid, among others, as the only carbon source. A *Burkholderia* sp. strain was isolated due to its distinct enhanced growth in these non-conventional substrates, assimilating them into oxalate and malonate. The genomic sequencing of the isolate showed the existence of genes coding for laccases and peroxidases that might become future targets for genetic improvement. The use of molecules representing polyaromatic structural diversity to screen the microorganisms led to three different potential catabolic pathways. Additionally, in this reference, there is a collection of quick testing protocols for activity using spectrophotometric scans.

## 4. Strategies for Advancing and Innovating Metabolic Engineering in Microorganisms for Applications

The naturally evolved metabolic pathways of microorganisms are the starting point for developing novel capacities of biotechnological interest. However, when one such potential microorganism is identified, several methodologies of molecular intervention must be started. For example, Pan et al. [80] looked at what had been done to make a solid molecular platform for *Cupriviadus necator*, a bacteria that could work in microbial electrosynthesis (MES), and proposed a technology in which microbes fix CO_2_ and electrochemical reactions provide redox equivalents. Currently, *C. necator* has vectors available for genetic engineering thanks to the systematic testing of replication origins, integration sequences (to avoid vector loss), promoters, auxiliary untranslated regions, signal peptides, and terminator sequences. Another relevant aspect is the selection of strains with desired capacities for molecular biology manipulation, such as transformation competence, selection marker compatibility, optimized codon usage, suitable natural mutagenic rates, and the development of stable mathematical models of metabolic flux. The research of all these factors allows the building of a solid microbial platform, such as *C. necator*, where at least a dozen industrial products have been synthesized. This exemplifies how far international efforts can go to advance microbial platforms.

An intriguing aspect of biomass conversion is how this objective is efficiently achieved in nature. Perez-Braga et al. [81] suggested that natural microbial consortia might share the load of different metabolic activities needed to move the whole process forward, probably by specializing in detoxification, cofactor production, and complementing pathways. In this direction, they determined several metagenomes from composting cells, the most common transcriptomic activities, and metabolite clusters to suggest modeled consortia relationships. The proposed interactions point out the importance of proton, ammonia, gas, and organic metabolite exchange. Although the apparent complexity observed may be overwhelming, this interesting approximation paves the way to proving several hypotheses of fine interactions among degrading consortia.

From this perspective, it is also interesting how a lignin-degrading microorganism deals with the diversity of molecules derived from this polymer. Studies of ecological niches that are rich in lignin show that many metabolic intermediates with different structural properties are made there [82]. Tan et al. [83] found lignin pathways and metabolic intermediates in *Streptomyces thermocarboxydus* by using an omics-based approach. In addition to traditional enzyme activities such as peroxidases and laccases, complete depolymerization stages, the effects of catabolizing the different chemical moieties, auxiliary pathways such as proton production, and the genomic description of operons were shown.

Even a single product demands the accumulation of extensive knowledge. Mezzina et al. [84] recently reviewed the state of the art of polyhydroxyalkanoates (PHA) production in *Pseudomonas putida*. PHA and its derivatives are good alternatives to plastics. Because they have evolved to be energy stores for cells, they are subject to strict metabolic controls that link several fundamental pathways, such as three different glycolytic pathways, aromatic catabolism, fatty acid metabolism, and TCA. The authors emphasized the future modification of substrate preferences to incorporate unconventional chemical moieties to create new technological applications of polymers.

Microbial biotechnology can also target desired niche markets such as levulinic acid, which is a by-product of the acid saccharification of plant biomass. Habe et al. [85] reviewed a very dynamic subset of research that deals with improvements in the utilization of levulinic acid (LA) using microbial cells that would allow advanced metabolic engineering to improve its usage. LA is a secondary product of lignocellulose hydrolysis that can be used by different bacterial species to yield bioplastic monomers and ketonic solvents. This includes the recently discovered catabolic pathways, its metabolic control, substrate preferences, operons, adequate NADH regeneration systems, and different pathways to direct its carbon to different reactions such as ligation, reduction, or decarboxylation.

The screening of natural diversity for microbial species of potential industrial use leads to candidates that can pass phenotypic and genetic analysis, a second level of characterization [86]. This has many advantages, such as the fact that having a high-quality sequenced genome accelerates the discovery of genetic determinants and opens the door to the creation of specific genetic tools for those microorganisms. In addition, it allows the proposal of solid mathematical models aimed at helping protein and pathway engineering, which is of primary interest. Balagurunathan et al. [87] provided an example of such experimental setups that accompany the genome-sequencing project of *Scheffersonmyces stipitis*, a microbe that shows preferential use of xylose. The use of predesigned phenotype arrays to test carbon, nitrogen, pH tolerance, and inhibitory substances, coupled with in-house macromolecular composition and computational modeling, allowed for the identification of the redox status of a cell as a crucial factor to improve xylose utilization.

Lignin used to be considered an economically unusable fraction to be used in biological methods. However, bacteria that can incorporate aromatic rings into their primary metabolism have been discovered. For example, Johnson and Beckham [88] compared two different pathways to incorporate benzoate or coumarate catabolism from *Sphingobium* into the industrially malleable *Pseudomonas putida* to incorporate carbon from lignin into PHA. First, the pyruvate dehydrogenase complex had to be knocked out by gene deletion to force carbon utilization from the aromatic molecules. Next, redundant pathways were incorporated separately, and it was found that some were sensitive to metabolic control, showing an extended lag phase, while others allowed for the immediate use of the lignin-derived molecules. This scheme opens up the possibility of using the abundant carbon biomass stored in lignin. This is a study case of how the knowledge of different pathways that lead to a desired industrial product is transferred into well-characterized organisms to expand their biocatalytic capabilities.

Thermophilic microbes have been investigated as biocatalysts for biomass valorization to directly act on industrial streams, avoid mesophyll contamination, and increase process control options. These organisms can also be subject to the recombinant incorporation of novel pathways. The use of artificial operons, i.e., multigenic tandems under the control of a single promoter, can be used to incorporate full pathways into the organisms of interest. For example, Keller et al. [89] cloned artificial operon genes from different bacteria to enable butanol production in *Pyrococcus furiosus* while simultaneously deleting (by insertion) the gene acetyl-CoA synthase to direct Ac-CoA towards butanol. The genes configuring these artificial operons were selected from different thermophyllic organisms to be used in *Pyrococcus furiosus*, which is an extremophile itself. The use of thermophyllic pathways would be an effective operative control to avoid mesophyllic microorganisms acting as carbon leakage into undesired products.

When an enzyme has been isolated and biochemically characterized, there are several strategies for improving its parameters. One is the rational approach, in which researchers look at the tertiary structure of the enzyme and decide to use site-directed mutagenesis in codons for amino acids that might change the size of pockets or control points. This scheme can be combined with evolution techniques applied only to sections of target proteins that have the potential to improve the whole enzyme. You et al. [90] used the structural knowledge of TIM-barrel proteins to observe that the structural loops of xylanase are where substrate recognition and catalysis take place. Saturating mutagenesis was applied in these loop sections, and, after screening, the kinetic parameters were improved and the thermotolerance was increased. By making the amino acid smaller or less reactive, the authors made a triple mutant that could work at a wider range of temperatures and had better catalytic parameters when tested on corn-based substrates. These improved mutants can be coupled in the future to adaptive laboratory evolution (ALE) for non-directed improvement.

The biomass intended to be bioconverted into products is usually treated to deconstruct its natural structure, which evolved to avoid the attack of microorganisms. These chemical pretreatments can produce secondary products that inhibit microbial growth. One example is furfural and its derivatives, which are abundant side-products of biomass hydrolysis. Guarnieri et al. [91] engineered a strain of *P. putida* to include furfural catabolism into the TCA cycle by recombining an eight-gene operon from *Burkholderia*. This allowed growth in both model substrates as well as in corn stover hydrolizates. This modified strain was subjected to serial adaptation to obtain a clone that was able to grow in aldehydes and could be further used as a chassis to build up metabolic controls, optimizing the catabolic capabilities and neutralizing and catabolizing such toxic molecules.

Cellulose and hemicellulose are the plant cell-wall polymers receiving the most attention with regard to their exploitation as bioconversion substrates. However, pectin and alginate are uronate-based polysaccharides abundant in plants and macroalgae, respectively. Pectin is a plant polymer that is usually considered a setback in plant biomass fermentation. Although microbes with pectin and alginate catabolic pathways have been known, there are still enzymatic steps missing. Hobbs et al. [92] reanalyzed the loci of different alginolytic bacteria to question the role of KdgF, an evolutionary-common open reading frame coding for an unannotated protein. Combining the characterization of the recombinant protein and its X-ray crystallographic analysis, it was defined as a catalytic tautomerase-like protein, increasing the uronate catabolism of pectinolytic microbes. In this way, KdgF activity and structure demonstrated that it is an enzyme that completes the full pathway of pectin and alginate catabolism. The recombinant enzyme from *Yersinia enterocolitica* was functional, and together with the previously known enzymes of uronate catabolism, it can be used to provide pectinolytic capacities to other microorganisms. This research proves that revisiting known pathways can provide opportunities to improve them in novel industrial contexts.

Diversity can also be manipulated at the source. Soils are a rich source of microbes with the metabolic capabilities of biomass deconstruction. Borjigin et al. [93] employed metagenomics methodologies to observe how the microbiome changed with different nitrogen treatments and if the cellulolytic activity changed. The research produced a catalog of microbes and their activities that enabled the conversion of agricultural waste into soil nutrients. Although the original context of this research was the agricultural practice of the exploitation of straw in soil restoration, it also proved the value of changing the nutrient balance in the original population to enrich interesting enzymatic activities for the industry.

Another approach to the problem would be to artificially create diversity in an already interesting strain, since complex polygenic characteristics are the hardest to improve. Therefore, the induction of genetic diversity through physical mutagens coupled with a growth in media with the desired selective pressures is a powerful tool. For example, *Candida intermedia* is a yeast that already has similar growth rates for xylose and glucose. Therefore, Moreno et al. [94] incorporated UV mutagenesis followed by ALE on wheat straw hydrolysates and ethanol as selective pressures. The experimental design allowed the recovery of yeasts with higher tolerance to inhibitory phenolic compounds commonly obtained as undesired side-products of lignin chemical processing. The new strain was further challenged to grow on xylose and increasing concentrations of ethanol to evolve tolerance to this end product. The subsequent molecular characterization of these strains will inform novel molecular mechanisms of carbohydrate and stress biochemistry.

The recalcitrance of lignin to biotransformation is the reason behind its high abundance and evolutionary conservation in nature. For that reason, the cost of its chemical deconstruction has to be recovered through the complete use of all the produced molecular components. Bioreactor research has identified the inhibition of growth by organic molecules present in hydrolizates as a drawback of industrial biomass conversion by microorganisms. These inhibitors include products from plant biomass chemical deconstruction, mainly phenolic compounds from lignin and furfural moieties from carbohydrates. To select for strains resistant to these chemicals, Xue et al. [95] created an artificial inhibitory mix formulated from the chemical characterization of plant biomass pretreatments, representing a large range of molecular diversity to select for positive carbohydrate consumption, cell growth, and ethanol yield. This inhibitory mix was used to challenge a library of *S. cerevisiae* mutants; those strains with above- or below-average survival were identified. This chemical genomics approach revealed that the absence of genes related to protein folding, Golgi transport, and the pyruvate dehydrogenase complex increased susceptibility to inhibitors. On the other hand, mutants of vesicle traffic control genes and fatty acid biosynthesis increased resistance. Both categories indicate the need for further research to fine-tune susceptibility to inhibitors by overexpressing and/or knocking out the discovered targets. This scenario points out a complex tolerance mechanism involving secondary metabolic products and the endomembrane system.

Other directed evolution techniques, such as error-prone PCR, can also be used to improve promising candidates for such cocktails. Cecchini et al. [96] screened thousands of candidates for an endoglucanase to obtain five promising improved sequences that were able to substitute the commercial versions. For the mutations behind this improvement, there was a common theme of them not being necessary close to the catalytic site but rather reinforcing the internal and external interactions, thus showing the value of random mutations.

A promising research line to be developed in the future is the creation of novel techniques for gene discovery, especially since sequenced genomes are released on a daily basis in databases. HamediRad et al. [97] set out to improve *S. cerevisiae*, the classic model and workhorse of industrial microbiology, to assimilate more xylose. The strategy involved the sense and anti-sense expression of cDNA libraries (named RNAi Assisted Genome Evolution, RAGE) coupled with size-based screening under the desired pressure. Since the expression was launched from the genome, the sequence causing the desired phenotype can be recovered from the integration site. The obtained clones indicated that vacuolar traffic (up-regulation) and cell division regulators (down-regulation) increased xylose catabolism and ethanol yield, opening up the possibility of studying nuclei–mitochondria–endomembrane relationships in a biotechnological context.

As reviewed in the examples herein provided, the isolation of the desired improvements involves the screening of thousands of candidates. Another technical advancement is the development of screening techniques that increase throughput and decrease costs. Sana et al. [98] designed an experimental strategy to isolate DNA promoters that respond to vanillin, a lignin catabolic byproduct used as a proxy for lignin metabolism. Through transcriptomic RNA-seq, the promoter yeiW was discovered and cloned in *E. coli* to direct the expression of the green fluorescent protein. This biosensor produced a fluorescence proportional to the amount of vanillin. We previously mentioned that lignin deconstruction produces dozens of products with high chemical diversity and individual catabolic pathways; it will be interesting to see this strategy applied to all of the individual pathways used for lignin catabolism.

The challenge of xylose catabolism can be solved using eukaryotic organisms too. Wohlbach et al. [99] used a mixed comparative genomics design based on bioinformatic screening and de novo DNA-sequencing projects to find promising xylose-catabolizing fungal species. This combination produced a set of enriched genes present in xylose fermenters, revealing signal peptide sequences, transporters, transcription factors, and unannotated genes, as well as the expected carbohydrate-metabolizing enzymes and accompanying redox regeneration pathways. The transgenic testing of these candidates probed their potential to increase the metabolism of this carbohydrate.

Biological catalysts have proven that their in vivo activities are constrained by the needs of the evolutionary history of the organism where they evolved. In vitro evolution and synthetic biology are techniques based on an evolutionary framework that uses the creation of diversity and subsequent screening to design new processes and products. Sherkhanov et al. [100] design a cell-free process that uses 16 enzymes to produce isobutanol. First, the enzymes were tested to tolerate high solvent concentrations; two of them required mutagenesis and further rounds of directed evolution to be sufficiently stable. A characteristic of this system is that one of its steps is NADP^+^ self-regeneration. As in other strategies, the authors stressed the importance of organic overlays to prevent product volatilization. These types of next-generation enzymatic reactors offer the promise of avoiding the need to stabilize a living system and instead working on the extended stability of the biocatalyst and its cofactor-regeneration systems.

The most extreme case of valorization is the use of CO_2_ produced as waste, and the construction of non-natural autotrophs is an attractive strategy. Autotrophy, the growth of cells using CO_2_, is widely distributed in nature. The transfer of this metabolic capacity into industrial heterotrophs would allow the removal of excess CO_2_ from the atmosphere and the exploitation of abundant residual CO_2_ streams. Gassler et al. [101] hypothesized that adaptive evolution would allow for an increase in the growth rate under CO_2_ assimilation. Another recombinant workhorse of yeasts, *Pichia pastoris,* has been transformed to possess the Calvin–Benson–Bassham Cycle (CBB), expressed at the peroxisome. Despite showing autotrophic growth, this strain was subject to an adaptive laboratory evolution to isolate mutants with increased autotrophy through the selection of non-rationally designed polymorphisms. Interestingly, one set of improved strains down-regulated the activity of the phosphoribulokinase Prk, one component of the recombinant CBB cycle. Another set of mutants was affected in the activity of nicotinic acid mononucleotide adenylyltransferase (NMA1) that would modulate the available NADH. This indicates the relevance of cofactor management in autotrophic growth. Despite that, both mutations, when incorporated together, did not potentiate the growth rate. A further adaptive evolution of NMA1 mutants revealed PEX5, a peroxisome signaling protein, as a crucial gene for further growth enhancement in autotrophic conditions. This research is paradigmatic in the sense that recombinant technology coupled with experiments designed for specific environmental pressures can lead to improvements through the mutation of unexpected targets.

## 5. Combined Molecular and Evolutionary Strategies

Sometimes, the desired bioconversions constitute residual activities of biocatalysts in nature. In that case, directed evolution can be an option to achieve a biotechnological result. Directed evolution takes the molecular concept of achieving a functional change in response to a changing environment, but genetic diversity is introduced through molecular biology, mostly (but not restricted to) PCR methods such as error-prone PCR or site-saturated mutagenesis, followed by an extensive selection of recombinant sequences.

Wang et al. [102] (2021) approached the technical challenge of using formate, a single-carbon organic acid, and incorporating it into the central metabolism. The simplest of the organic acids is a common subproduct of the chemical deconstruction of biomass. For this, the authors started with acyl-CoA synthetase and acylating aldehyde dehydrogenase, enzymes with residual activity on formate, and through site-saturated mutagenesis, diversified the coding sequences surrounding the active site. The objective was to increase the substrate preference from acetate, the prevailing two-carbon acid in cells, to formate. The subsequent enzymatic scrutiny indicated recombinants with improved use of formate, not through increased enzymatic parameters but with improved soluble expression and decreased preference for acetate. Disappointingly, the use of both improved enzymes to incorporate formate into the central metabolism of *Methylobacillus flagellatus* could not be observed in vivo, indicating a research opportunity to find and overcome the poor redirection of formate to the central metabolism, most likely by outcompeting mineralizing pathways. This research laid the foundation to twitching the metabolic pathways toward the goal of monocarbon acid assimilation.

Ethylene glycol (EG) is usually associated with the synthetic plastic PET; however, it is also a product of syngas that may be produced from biomass. *Pseudomonas* can use EG as a carbon source, but *Paracoccus* has a more energetically efficient pathway: the B-hydroxyaspartate cycle (BHAC). Using *E. coli* as an optimizing intermediate, von Borzyskowski et al. [103] expressed BHAC in *Pseudomonas* and performed proteomic determinations to observe that the introduction of this pathway induced a full rearrangement of carbon metabolism. ALE was further applied to improve the process and registered a complex scenario where export systems, protein sensors, and transcriptional regulators were mutated.

A common issue in combined fermentation is the genetically coded preference for metabolizing the diverse available carbohydrates with different priorities. Generally, glucose is consumed first, followed later by other sugar moieties such as xylose. This issue is not resolved only by promoting xylose-assimilating enzymes because the genetic circuits controlling the preference are multigenic. These hierarchies hold back the use of hemicellulose. Therefore, the study of classic catabolic pathways and their rational intervention is limited in its ability to achieve complex phenotypes. For this reason, Kim et al. [104] approached this challenge in *E. coli* through the theoretical modeling of metabolic flux and the experimental knockout of genes coupled with adaptive evolution. The modeling indicated that the deletion of phosphoglucosiomerase and GntR (a transcriptional repressor) would partially block glucose use and force the metabolism to employ xylose. When experimentally tested, it allowed the concomitant catabolism of both sugars. Surprisingly, a population evolved that reverted to the wild-type phenotype; its full genome sequence revealed mutations that increased glucose utilization and suggested further deletions on these targets. These strains were also subjected to glucose/xylose co-utilization and ethanol resistance through adaptive evolved rounds of cultivation. The whole-genome sequencing of evolved strains showed the mutation of transcriptional repressors and gene promoters and a complex scenario involving the differential use of transporters, the altered expression of the glycolytic enzymes, and the phosphotransferase system. Evolved strains grew faster in both the model and the complex switchgrass hydrolysates.

Another example is the work by Kao et al. [105], where a glucosidase from the fungus *Chaetomella raphigera* was subjected to both rational site-directed mutagenesis from the active site and error-prone PCR to create libraries. When improved versions were sequence-analyzed, it was found that the mutations fell outside the active site and may enhance activity by improving solubility and decreasing cellular stress caused by recombinant enzyme production using *Pichia pastoris* as a testing platform. Some mutations were proposed to improve protein stability and folding in the endoplasmic reticulum. This research highlighted the importance of incorporating non-rational strategies into the protein engineering toolbox to achieve results otherwise improbable to predict.

In addition to making available new strains for the industry, studies involving an evolution-based set of techniques also produce new information about unexpected metabolic relationships. Sato et al. [59] applied to yeast a selective pressure of minimum glucose and high xylose media, followed by subsequent rounds of anaerobic pressure. The strain used as the parental strain had recombinant expression of xylose catabolic genes. The collection of obtained mutants increased xylose consumption and revealed a large set of mechanisms leading to adaptations, such as signaling, Fe-S cluster biosynthesis, and oxidative stress prevention. The combinations of mutations, although not always yielding positive biotechnological results, revealed agonistic and antagonistic relations among pathways. With the use of RNA-seq, proteomic, and metabolomic tools, an expression atlas of several mutants and their combinations was produced.

Innovation in plant biomass pretreatment has included the use of ionic liquid solvents. Unfortunately, these chemicals commonly inhibit microbes. Higgins et al. [106] isolated strains from composting biomass that were tolerant to these types of solvents through gradual exposure to them. The isolates belonged to the *Bacillus* family and were further used as donor genomes to find sequences conferring this tolerance on *E. coli* non-tolerant cells. Fosmid library screening and sequencing indicated that small multidrug resistance (SMR) proteins were the genetic determinants of enhanced strains and that this could be used to create recombinant strains of other genera. Further genetic comparisons indicated mutations in stem-loops of transcripts likely enhancing the translation of mRNAs in response to guanidine, a toxic ionic natural metabolite.

## 6. Strain Improvement by Characterization of Integral Control of Metabolism

The isolation of microbes with multiple metabolic activities of interest for biomass valorization from effluents or spent substrates is a classic research approach. Li et al. [107] took this method and went one step further by incorporating a glucose analog, 2-DG, into the selection. When 2-DG is metabolized, an intracellular inhibitor is produced, arresting growth unless the cell is insensitive to catabolite genetic control. In this way, the authors were able to isolate a new strain of *Thermoanaerobacterium thermosaccharolyticum* with silenced catabolite repressor genes (Ccr and XylR), enhanced extracellular export proteins, and tryptophan biosynthetic genes. The whole transcriptome analysis of the new strain indicated other natural factors, such as improved NADH generation and the absence of pathways leading to unwanted secondary fermentation products such as acetone. The authors propose that, taken as a whole, these interrelated aspects may indicate the importance of tryptophan as a precursor of NADH/NADPH biosynthesis, directly impacting glycolysis and fermentation. This work pointed out alternative improvement nodes, not necessarily involving classic metabolic enzymes but deeper levels of genetic and biochemical control.

Cheng et al. [108] explored the fundamental biochemistry of *E. coli* strains with increased glycerol uptake and mutations in genes such as glycerol kinase, RNA polymerase, and their combinations. An interesting approach was merging the classic measurement of internal metabolites with exometabolites and their contrast. This allowed for the observation of the role of carbon wasting in maintaining growth rates and increasing fermentative products. Finally, the importance of cofactor replenishment and ratios as consequences of different mutations was highlighted.

Another example where carbon waste was researched is that of *S. cerevisiae*. In most cases reviewed, ethanol is the desired final product of fermentation. But if the intended product is not ethanol but other qualities of yeast used in industry are wanted (such as tolerance to shear force, aerobic growth, and a fast rate of doubling), carbon mobilization toward fermentation must be stopped. For that purpose, several metabolic engineering steps had to be taken, such as the substitution of pyruvate dehydrogenase complex enzymes and the deletion of acetate pathways. Yao et al. [109] performed a transcriptomic analysis on this strain to observe the pathways most likely to be receiving the new flow of carbon to design the production of new non-ethanolic NADH-dependent final products such as lycopene, fatty acids, and 2,3-butanediol.

Well-established industrial enzymes can be further improved through directed evolution methodologies. Goedegebuur et al. [110] aimed to improve the thermotolerance of Cel7A, a cellobiohydrolase, through site-specific combinatorial mutagenesis. The most thermostable version was obtained through the accumulation of up to 18 mutations, many isolated individually, and, importantly, they would not have been discovered through natural screening since the mutations are not present in reported sequences. It was hypothesized that the induction of hydrophobic interactions and the rigidity of loops may be the molecular mechanisms that sustained the improvements.

Plastics are commodities with a high environmental impact but play a crucial role in modern society; for example, nylon and all the associated industries around it. Strains of *Pseudomonas putida* can direct aromatic rings from lignin to muconate, a monomer for nylon substitutes. Based on previous observations that aromatic catabolism is repressed by genetic controls, Johnson et al. [111] decided to knockout *crc*, a protein acting as a translation repressor of mRNAs coding for enzymes in biochemical pathways and transcription factors. The recombined strain was able to preferentially catabolize the aromatic rings and direct the carbon flux to muconate, the desired product, without large observable secondary effects on the cell.

With the advent of next-generation sequencing technologies, the cost of sequencing has been steadily decreasing while the sequencing capacity has been increasing. The field of biomass conversion has benefited from such technologies to understand the complexity of substrate preference and the genetic control surrounding it. When genomic information is available, it acts as a blueprint where the pathways expressed under biomass-derived substrates can be traced in an experimental manner. Rodionov et al. [112] determined, through transcriptome sequencing (RNA-seq), the expressed genes of thermophilic bacteria *Caldicellulosiruptor bescii* when growing on substrates of different degrees of complexity (from glucose to xylan), representing the main carbon sources obtained from plant biomass deconstruction. This allowed the assembly of exoenzymes, transporters, activators, and repressors, as well as biochemical pathways, either common or specific to the tested substrates, as well as the construction of regulons, their promoters, and regulatory DNA sequences. To test the depth of the research, a knock-out mutant of a detected ATPase component (msmK) demonstrated that this gene serves transporters and is essential to allow growth on polymeric plant substrates. This ATPase seems to couple to transmembrane transporters, allowing the import of di- and oligosaccharides, and highlights the potential to further optimize this step. Such studies are a source of novel genetic candidates to mutagenize, edit, and recombine for the improvement of the various operations needed for biomass conversion in an economically feasible manner. It can be predicted that the metabolic landscape will be expanded from individual enzymatic activities to incorporate transporters, activators, repressors, and control checkpoints.

Engineering metabolic flux is now more feasible than ever. Thanks to genomic projects and increasing metabolomic analysis, coupled with RNA sequencing and computing power, the techniques of adaptive laboratory evolution have the power to deliver extensive knowledge and technological applications. This was applied to the problem that plant cell walls are the main component of agricultural waste that can be bioconverted into trade chemicals, and an ideal bioprocess should use all the different carbon structures present. Elmore et al. [113] deleted the gene for glucose dehydrogenase (gcd) for less glucose metabolism and by-products. Then, genes from the pentose phosphate pathway and xylose transport were incorporated. Strikingly, subsequent rounds of adaptive evolution allowed for the finding of mutations in the promoter regions of the recombinant xylose transporter, indicating that transport was withholding the usage of xylose. Additional arabinose catabolism pathways were incorporated, thus making available a single organism suited to employ multiple carbon structures present in real plant hydrolizates.

Transforming biomass into industrial chemicals can be directed through recombinant strategies and further improved through adaptive evolution. For example, the cellulolytic bacteria *Clostridium thermocellum* has been rationally engineered to produce lactic acid, a valuable industrial product. However, the use of cellulolytic microorganisms in industrial setups needs costly measures to avoid their inhibition by acidic pH. These strains now need to be improved in their acid tolerance. For this purpose, Mazzoli et al. [114] employed adaptive evolution in the presence of high concentrations of sodium lactate, alternated with lower selective pressure periods. In this way, authors were able to isolate different strains with improved lactate tolerance and determine that the mutated genes included retained knockout mutations in phosphoenolpyruvate kinase (PPDK), insertions in the metabolic regulatory protein histidine kinase HPrK/P, mutations in DNA repair mechanisms, and genes of an unknown function. Surprisingly, these mutations indicate a change in carbohydrate metabolism rather than a direct chemical detoxification mechanism. Nonetheless, other mutations were found, but on genes that lack structural or functional information. Despite these apparent drawbacks, it is important to highlight that these results could not have been obtained by rational design and, in consequence, open unexplored avenues of biochemical knowledge.

The knowledge of synthetic biology and metabolic flux has been used to achieve autotrophic CO_2_ fixation in *E. coli*. Gleizer et al. [115] designed an initial microorganism that expressed Rubisco and auxiliary enzymes such as carbonic anhydrase. In addition, it had a set of deleted central metabolic nodes of carbohydrate catabolic pathways to force carbon channeling through the incorporated autotrophic network. Through ALE, this initial strain was subjected to decreasing external sugar feeding to increase its reliance on autotrophic growth. NADH was supplied through transgenic formate dehydrogenase. The mutations accumulated in the evolved strain showed the down-regulation or deletion of enzymes that would allow the escape of fixed carbon to other pathways such as pentoses, trioses, and amino acid metabolism. This work presents, in a radical phenotype, the ability of gene networks to shape survival in extreme, unnatural conditions in relatively few generations (~150).

The improvement of metabolic characteristics is achievable not only in single organisms but also at the community level. This concept can be applied to enrich multiple genotypes that can serve as a natural library for subsequent improvements. Ceron-Chafla et al. [116] used adaptive laboratory evolution to improve cell growth under increased partial pressures of CO_2_. When CO_2_ is dissolved in water, carbonate is produced. This molecule can in turn be toxic for bacteria, creating an obstacle to the waste conversion to methane, especially when using anaerobic conditions. The proportion of bacterial diversity was negatively changed by the applied selective pressures, indicating the successful discard of non-adapted genomes. The consequence of this selection was an increase in biomass and methane production and the selection of genotypes with yet-to-be characterized metabolic activities. The evolved consortium was enriched with archaeal species that can reveal, through subsequent experimentation, the molecular mechanisms of tolerance to high pCO_2_.

Glucose is one of the main metabolites released from biomass deconstruction and is further used for chemical bioconversion. In natural conditions, glucose can act as an inhibitor of other pathways, such as xylose catabolism, that are of industrial interest and need to be simultaneously active. To increase the economic feasibility of plant biomass conversion, all molecular fractions of hydrolysates should be suitable as substrates. Therefore, Kurgan et al. [117] tested the capacity of major facilitator proteins (MFS) to be adapted through laboratory evolution to increase the diffusion of xylose into the cells. Direct evolution was used to increase xylose transport by Glf, a transmembrane sugar uniporter. For this purpose, an *E. coli* strain genetically disabled for glucose and xylose uptake was used as a background. Glf was subjected to directed evolution by error-prone PCR and mutagenesis, and a set of new Glf-derived proteins with more xylose transport capacity were obtained. These evolved proteins contained mutations in the sugar-binding amino acids that weakened glucose inhibition, as well as other mutations in amino acids that were not obviously associated with the known mechanisms of transport. This is another step in the construction of strains capable of industrial xylose utilization, in parallel with the elimination of carbon catabolite repression (CCR) by mutating xylR into a glucose-insensitive version and achieving furfural tolerance or catabolism.

Directed evolution can also reduce CCR. Sievert et al. [118] subjected *E. coli* cells to a minimum medium with a high xylose concentration (10%) and recovered strains with mutated CRP and XylR transcriptional activators of the xylose catabolic operons. In addition, transposition events (IS10) were detected on a formate channel (FocA) that negatively impacted the transcription of pflB, a neighboring gene involved in increasing the fermentation rate. The authors contrasted the mutations accumulated in their strains with others and highlighted how different experimental settings produce different solutions to the applied pressure. In addition, the serendipity of the solutions highlights the importance of evolutionary methodologies.

The inhibition of bacterial growth by secondary products from biomass chemical deconstruction, such as furfural, is a pivotal research area to achieve full carbon usage. Zou et al. [119] used ALE with furfural as selective pressure on a previously engineered *Pseudomonas* strain to develop tolerance. The sequencing of the evolved strains revealed that organic molecules transporters of the ABC family accumulated the mutations and could be further used to increase tolerance in a recombinant manner.

## 7. Non-Conventional Wastes and Auxiliary Pathways

Lignocellulosic conversion has led the way in the research of substrates for biofuels. However, microorganisms can also be tailored to catabolize other residues to solve industrial pollution and help create a circular economy. Rodríguez et al. [120] tackled slaughterhouse waste by incorporating in *Cupriavidus necator* a two-enzyme lipolytic pathway. This microorganism was selected because it is a producer of polyhydroxyalkanoate (PHA), a substitute for synthetic plastic monomers. This straightforward strategy achieved the desired carbon-channeling pathway using different residues and set the fundamentals to optimize the concept on a large scale.

The transformation of biomass can be directed toward specialty chemicals. Cai et al. [121] researched the production of the pharmaceutical reagent (S)-4-aminopenanoic acid from levulinic acid, a chemical produced through corn biomass chemical deconstruction. For that purpose, an improved amine dehydrogenase was evolved from its basal activity. The first round of evolution (error-prone PCR) improved solubility, while advanced rounds impacted enzyme activity. Finally, the complete process was demonstrated, from starch acid hydrolizate to the chiral product.

The knowledge of biomass conversion can be extended to the conversion of artificial waste. Tan et al. [122] showed how a screening using aromatic oxidation as biological pressure could be used to separate PHA-producing strains using styrene as a carbon source. These isolates are native and therefore not yet engineered for the optimal use of artificial substrates, but once the basal activity is obtained, they can be improved through the high-throughput methods used for other substrates herein reviewed.

Some biological catalysts are also dependent on metal ions acting as prosthetic groups. However, cells have mechanisms to control the cytoplasmic metal homeostasis that may deprive enzymes of their full catalytic potential. Lima et al. [123] reasoned that eliminating vacuolar proteins of metal ion homeostasis would increase ethanol production from xylose. Although counterintuitive at first sight, the mutants could accumulate more Fe and Mn in the cytoplasm and increase ethanol production. Surprisingly, no pleiotropic effects were recorded on ROS management or growth derived from the increase of cytoplasmic metal ions. When the transcriptome of mutants was measured, each mutant displayed its own expression adjustments in a wide range of cytoplasmic and organelle pathways, highlighting the interplay between metal homeostasis and carbon metabolism.

Other aspects deserve research interest, for example, microelement nutrition. Methanotrophic bacteria can use this C1 molecule to produce biofuels or other chemicals. Interestingly, it was observed that lanthanum could affect the gene expression of methanotrophic bacteria, with consequences for both the core metabolic routes (i.e., TCA) and methane-specific metabolizing enzymes [124]. The preference for calcium and lanthanum indicates an intriguing evolutionary history in the natural niches of these bacteria.

Other specialty chemicals can be additives for combustion engines. Yuzawa et al. [125] proposed the use of short-chain ketones as gasoline oxygenates. In this work, the polyketide synthase family of enzymes was engineered through the modular study of domains. When expressed in *Streptomyces* strains, these hybrid enzymes allowed the use of corn stover hydrolyzates to produce six different methyl ketones. It was observed that the supplementation of amino acids improved the yield, proposing the concomitant production of such amino acids (e.g., isoleucine) as the next improvement step.

Plant biomass deconstruction has traditionally been directed towards soluble metabolites; however, it can also be metabolically channeled towards oleochemicals. Valencia et al. [126] aimed to increase the production of fatty acids in *Pseudomonas putida* by producing strains with knocked-out CoA-ligases to concentrate fatty acids and expressing methyltransferases to select for products of different chain lengths.

## 8. A Note on Safety to Achieve the Transition from the Laboratory into the Industry

New molecular and genetic tools have enabled the development of novel metabolic engineering and reduced the time and effort required to achieve high yields for novel biofuels and by-products. However, the developments in fermentation engineering have to be strengthened to avoid the risk of harmful environmental impacts due to leaks of the genetically modified microorganisms. This is an important part of having a successful transition from laboratory to commercial scales, and biosafety regulations may prevent the unwanted release of genetically modified microorganisms to the surrounding environment. This area of research involves extensive experimentation on genetic containment strategies such as preventing non-wild-type DNA (artificial or recombinant) from propagating, restricting the non-wild-type DNA in the external environment, and suicide loops. We refer the reader to the review by Stirling and Silver [127] that covers this ingenious molecular biology research. We understand that this adds an extra layer of complexity to the already vast horizon of microbial engineering possibilities. However, those technologies close to maturity and tested in all scale-up formats will probably be regulated in the future with these schemes of containment.

Therefore, it is crucial to clarify and regulate the legal framework for their application [128]. This requires ensuring the safe and viable use of genetic engineering techniques and conducting further scientific research to minimize the environmental and human health risks associated with them. The management of the residues generated, appropriate disposal, and circularization of these residues are also subjects of discussion in the field of global legislation.

## 9. Prospects of Engineered Microorganisms to Produce Biomass-Based Added-Value Compounds

Modern society requires a significant amount of energy to meet its demands. However, this energy consumption is facing numerous challenges that are affecting the environment, such as the depletion of resources, climate change due to the release of pollutants, food insecurity, and the excessive use of agrochemicals. These challenges have led to new strategies for producing fuels from various sources, including the development of biomass-based fuels using genetically modified microorganisms. These biofuels and by-products are categorized as fourth-generation biofuels (FGB) and offer potential technical and economic benefits.

While some genetic engineering tools have been shown to reduce production costs and improve the yields and quality of biofuels, they are still expensive and can only be implemented with global collaboration. Facilitating global trade agreements requires achieving global consensus, mediating potential conflicts, moving beyond technical studies, and focusing on the requirements for successful implementation.

One example of such state-of-the-art collaborative efforts is the fusion of the fields of molecular simulation and bioprocess design. Although this has been done previously for chemical engineering, bioprocesses pose a higher level of complexity because of the multiple scales of increasing organization typically involved in creating biotic structures. A case of this is lignocellulosic biomass, where molecular organization starts at glucose-monomer bonding and scales up to multimolecular layers and the directionality of vessels [129]. The basic parameters for mathematical models can be obtained through electronic microscope observations of enzymatically or chemically digested material, and multiple parameters for the process design can be calculated, such as surface and pore diffusion, hydrophobicity, effects of monomer composition, and, in consequence, recalcitrance [130].

Although the methodologies involved require a lot of computing power and advanced mathematical methods, researchers are working to make these pipelines more efficient and accessible, including through open-access repositories [131]. It is expected that sharing experience, building knowledge, and sharing computational resources can lead to an understanding of the molecular foundations of biomass recalcitrance, feedstock quality, possible modifications, and, most interestingly, a new angle for enzyme optimization. One study case is that of Vermaas et al. [132], where cellulose fibrils were modeled as hydrophobic and hydrophilic faces to observe how different lignin components can bind and form the basic structure of the secondary cell wall. The results indicated that the strength of the binding depends on the methoxylation degree of the lignin monomers. This work scheme has been applied to study the role of solvents in pretreating biomass for further deconstruction steps. The results allowed for the classification of nine different solvents into three classes with different effects on lignin and diverse monomeric compositions [133].

This knowledge is the basis for the proposal of what has been called “designer lignins” [134]. Since the experimental observations indicate that the arrangement and proportion of lignin monomeric units have large effects on biomass recalcitrance, plant cell walls can be biologically modified and optimized in vivo by modifying the lignin biochemical pathways. This has been achieved through different plant molecular biotechnology strategies such as transcription factor overexpression, cell-specific expression, and gene knockouts [2,134]. However, by incorporating a molecular simulation, the development of optimized enzymes for specific designer lignins can be achieved. The concept can also be applied to investigate the interaction of carbohydrate polymers such as cellulose and hemicellulose. Kisahni et al. [135] ran simulations of branched hemicellulose units and described the role of water displacement between the interfaces of the polymers as an important bioprocess design parameter. We expect that this field of research can be a bridge to join catalyst design efforts in the lab with the polymeric molecular diversity of plant biomass.

Here, we have reviewed bioengineering developments resulting in strains tested under controlled conditions that have the potential to be commercialized and transferred into the industrial sector (Figure 1). Reviewing the scientific literature is a good strategy to obtain cutting-edge tendencies in research, but for industrial sector advancements, patents may be a better indicator of these. Appendix A shows a compilation that can be used as a starting point for researchers aiming for the intellectual protection of their advances.

## 10. Conclusions

Advanced technological methods have been developed to improve the microbial metabolic performance in the bioconversion of biomass into valuable bioproducts. One of the most promising biotechnologies is the incorporation of evolutionary principles into the metabolic engineering of microorganisms such as bacteria, fungi, yeasts, and microalgae. This alternative has significant advantages over traditional methods for changing complete metabolic pathways or optimizing auxiliary pathways parallel to recombined genes. It improves the ability of microorganisms to bioconvert, and it gets around problems such as needing a specific substrate or being stopped by unwanted compounds. With the increasing demand for biofuels, the use of engineered microorganisms is becoming essential, leading to the growth of a new biotechnological fuel industry. Finally, the prospects of engineered microorganisms to produce biomass-based high-added-value products are bright, and with the abating prices of sequencing, the most daring experimental designs that used to limit their potential can now be performed. These engineered microorganisms have the potential to improve the efficiency of bioprocesses and contribute to sustainability efforts.

## Figures and Tables

**Figure 1 microorganisms-11-02197-f001:**
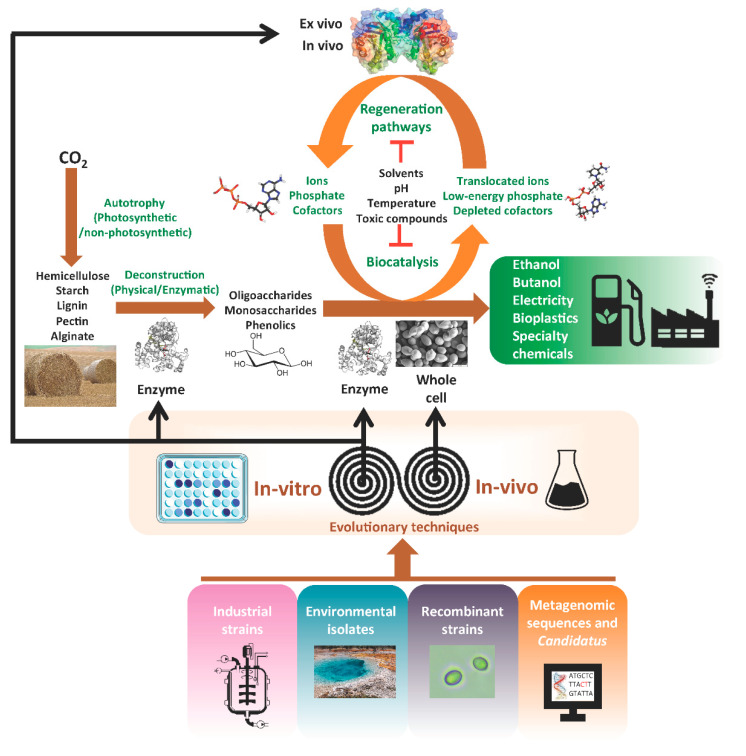
Overview of the modern available strategies for engineering the metabolic landscape of microorganisms for the production of biofuels and other chemicals. Illustrations are original or under the Creative Commons licenses of the following users: YassineMrabet, Dietmar Rabich, Yikrazuul, Daniel Plazanet, Thomas Shafee, Shaojie Yang, and Mogana Das Murtey.

**Table 1 microorganisms-11-02197-t001:** Metabolically engineered microorganisms for the conversion of cellulosic substrates into high-value bioproducts4. Modern screening of natural diversity to obtain catalytic capacities.

Microorganism		Substrate	Product	Pathway	Improvement of Modified Strain	References
Bacteria						
	*Klebsiella pneumoniae*	Glucose	2-Butanol	Meso-2,3-butanediol synthesis	320 mg/L of 2-butanol 720 mg/L by knocking the IdhA gene and adding coenzyme B12 1030 mg/L by engineering the diol hydratase	[44]
	*Escherichia coli*	Synthetic medium	Fatty alcohol	Fatty acyl-ACP reductase-dependent	0.75 g/L of fatty alcohol	[45]
	*Corynebacterium glutamicum*	Glucose, xylose	3-hydroxypropionic (HP) acid	Glycerol	62.6 g/L 3-HP	[46]
	*Clostridium acetobutylicum* and *Saccharomyces cerevisiae* (yeast)	Glucose, corn, corn stover, and starch	n-Butanol	Clostridial acetoacetyl-CoA-derived pathway	16.3 g/L of butanol	[47]
	*Corynebacterium glutamicum*	Glycerol	1,3-propanediol (1,3-PDO)	Glutamate fermentation	Conversion of glycerol into 1,3-PDO of 1.0 mol/mol glycerol	[48]
	*Klebsiella pneumoniae*	Glucose	2-butanone	2, 3-butanediol synthesis pathway	450 mg/L of 2-butanone	[49]
	*Escherichia coli*	Glycerol	3-hydroxypropionic acid (3-HP)	Expression of *dhaB* and *aldH*	31 g/L of 3-HP	[50]
	*Escherichia coli*	Glucose	3-hydroxypropionic acid (3-HP)	Modulation of malonyl-CoA reductase (MCR) activity	40.6 g/L of 3-HP	
Yeast						
	*Saccharomyces cerevisiae* strain XUSAE57	Xylose and Glucose	Ethanol	Xylose-isomerase pathway	0.43–0.50 g ethanol/g xylose	[51]
	*Saccharomyces cerevisiae*	Raw corn or casaba starch	Ethanol	Glucoamylase expression	Better fermentation performance, observing a reduction of at least 40% in the dose of glucoamylase	[52]
	*Saccharomyces cerevisiae*	Sucrose	Ethanol	Modification to resist chemical stress	Increased the ability to resist stress factors by changing the cell membrane components, expressing transcriptional regulatory factors, activating the anti-stress metabolic pathway, and enhancing ROS scavenging ability	[53]
	*Saccharomyces cerevisiae*	Non-glucose sugars and cellulose	Ethanol	Xylose isomerase and 1-epimerase expression	Improved cellobiose utilization	[54]
	*Saccharomyces cerevisiae*	Cellulose	Ethanol	Xylose isomerase expression	It was found to improve ethanol production from non-detoxified hemicellulosic hydrolysates	[55]
	*Scheffersomyces stipitis*	Glucose, xylose, arabinose	Ethanol	Xylose reductase	Improvements of xylose fermentation on lignocellulose, showing defects in glucose catabolite repression and are more resistance to inhibitors present in hydrolysates	[56]
	*Saccharomyces cerevisiae*	Corn cob hemicellulosic hydrolysate	Ethanol	Introduction of Xylose isomerase (XI) and xylose reductase/xylitol dehydrogenase (*XR/XDH*) pathways	High ethanol productivities and yields from xylose	[55]
	*Saccharomyces cerevisiae*	Lignocellulose hydrolysates of *Arundo donax*	Ethanol	Expression cassette containing 13 genes including Clostridium phytofermentans *XylA*, encoding D-xylose isomerase (XI), and enzymes of the pentose phosphate pathway	Increased ethanol titer of 5.8% (*v*/*v*)	[57]
	*Saccharomyces cerevisiae* TMB3400	Xylose	Ethanol	Pentose fermentation XR/XDH (*S. stipitis XYL1*, *XYL2*) by *SsXYL1*, *SsXYL2 + XKS1*↑, random mutagenesis	Increased ethanol values up to 0.33 g/g	[58]
	*Saccharomyces cerevisiae* GLBRCY87	Glucose and xylose	Ethanol	*SsXYL1*, *SsXYL2*, *SsXYL3*, evolved on xylose and hydrolysate inhibitors	Increased ethanol values up to 0.34 g/g	[59]
Microalgae						
	*Scenedesmus quadricauda*	Beijerinck medium	Triacylglycerols (TAGs)	Elevating intracellular malonyl-CoA and glycerol-3-phosphate (G3P) by overexpression of Acetyl-CoA carboxylase (ACCase) genes	It was evaluated in *S. quadricauda* LWG002611 which exhibits high biomass as well as high lipid productivity, to improve it via molecular engineering	[60]
	*Phaeodactylum tricornutum*	F/2 medium without Na_2_SiO_3_·9H_2_O	Improvement in lipid accumulation	Overexpression of lysophosphatidic acid acyltransferases (LPAATases)	Increase of 1.81-fold in polyunsaturated fatty acids	[61]
	*Scenedesmus obliquus*	BG11 medium	Improvement in lipid content	Overexpression of the type 2 diacylglycerol acyltransferase (DGAT)	127.8%, 20.0%, and 232.6% higher production of lipid content, biomass concentration, and biomass productivity, respectively, compared to wild-type strain	[62]
	*Nannochloropsis oceanica*	F/2 liquid medium or F/2 agar plates	Improvement in triacylglycerol	Overexpression of the diacylglycerol acyltransferase (DGAT)	Increase of 69% in neutral lipid content	[63]
	*Phaeodactylum tricornutum*	Algal and humus media	Improvement in total lipids	Deletion of multifunctional lipase	Rapid increase in free fatty acid (FFA) content in engineered microalgae	[64]
	*Nannochloropsis salina*	F2N medium	Fatty acids	Heterologous expression of AP2 type TF Wrinkled1 of *Arabidopsis (AtWRI1* TF)	Increase in total lipid contents of 44.7%	[41]
	*Nannochloropsis oceanica*	F/2 medium	Fatty acids	Construction and overexpression of multiple fatty acid desaturases (FAD) in *N. oceanica* CCM P1779	Enhanced ω 3 long-chain polyunsaturated fatty acids (LC-PUFAs) and eicosapentaenoic acid (EPA) production	[65]
	*Synechococcus* sp.	BG11 medium	Short chain fatty acids (SCFAs)	Antisense expression of Synpcc7942_0537 (*fabB/F*) and Synpcc7942_1455 (*fabH*)	Fatty acid composition analysis showed C14 increased by 65.19% and 130%, respectively, when *fabB/F* and *fabH* were antisense expressed.	[66]
	*Mychonastes afer*	BG-11 medium	Lipids	Cloning and expression of 3-ketoacyl-coA synthase gene from *M. afer* (MaKCS) in *Saccharomyces cerevisiae* BY4741	Increased lipid content, especially nervonic acid, under stress conditions of high light and low nitrogen.	[67]
	*Phaeodactylum tricornutum*	f/2 medium	Lipids	Overexpression of glycerol-3-phosphate acyltransferase (GPAT1) and lysophosphatidic acid acyltransferase (LPAT1)	Increase in photosynthetic activity and lipid content without compromising growth	[68]

## Data Availability

Data sharing is not applicable to this article.

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
