# Peer review of "Engineering the Metabolic Landscape of Microorganisms for Lignocellulosic Conversion"

_microorganisms, 2023, doi:10.3390/microorganisms11092197_

Round 1

Reviewer 1 Report

To make the review more rigorous and comprehensive the authors must consider citing both experimental and computational work related to this issue. For instance, a large volume of computational work exists to understand and predict lignocellulosic behavior in different solvents and suggest the molecular origins of biomass recalcitrance. I would like to learn more about how the authors interpret the use of results in the context of the preview. Few examples to consider are:

1. https://doi.org/10.1016/j.jcis.2020.12.113

2. https://doi.org/10.1016/j.copbio.2015.10.009

3. https://doi.org/10.1021/acssuschemeng.9b07415

4. https://doi.org/10.1021/acs.energyfuels.1c02163

5. https://doi.org/10.1021/acssuschemeng.9b04648

6. https://doi.org/10.1016/j.ces.2023.118587

No comment.

Author Response

Answers to Reviewers

* Reviewer 1

To make the review more rigorous and comprehensive the authors must consider citing both experimental and computational work related to this issue. For instance, a large volume of computational work exists to understand and predict lignocellulosic behavior in different solvents and suggest the molecular origins of biomass recalcitrance. I would like to learn more about how the authors interpret the use of results in the context of the preview. Few examples to consider are:

  1. https://doi.org/10.1016/j.jcis.2020.12.113
  2. https://doi.org/10.1016/j.copbio.2015.10.009
  3. https://doi.org/10.1021/acssuschemeng.9b07415
  4. https://doi.org/10.1021/acs.energyfuels.1c02163
  5. https://doi.org/10.1021/acssuschemeng.9b04648
  6. https://doi.org/10.1016/j.ces.2023.118587

Response: We have included a discussion on the recommended references as an example of interdisciplinary and collaborative work in the Prospects section.

Reviewer 2 Report

1. Introduction – In my opinion the content of the Introduction chapter is sufficient. The chapter ends with a correctly formulated aim of the work, which is its advantage.

2. Traditional techniques for converting lignocellulosic residues into high-value products –

Lines 90 – 92 - please provide examples of parameters of the mentioned processes, e.g. acid concentration, duration of action, what enzymes, etc.

3. Enhancing biomass-based products through metabolic engineering

Table 1 - In this form, the table adds nothing to the manuscript. Please add two additional columns - one with the product yield of the wild strains and the other with the yield of the modified strains.

4. Modern screening of natural diversity to obtain catalytic capacities

Sugar-based biomass has received the most attention in biological conversion to chemical products - The description of this paragraph is insufficient - please provide more studies and examples.

The table that is now in supplementary materials should be included in the manuscript

The article is well written, with minor corrections it can be published in the journal Microorganisms.

Author Response

Reviewer 2

  1. Introduction – In my opinion the content of the Introduction chapter is sufficient. The chapter ends with a correctly formulated aim of the work, which is its advantage.
  2. Traditional techniques for converting lignocellulosic residues into high-value products –

Lines 90 – 92 - please provide examples of parameters of the mentioned processes, e.g. acid concentration, duration of action, what enzymes, etc.

Response: We have incorporated the reviewer's comment as an expansion of the paragraph to provide the readership with more engineering context.

Reviewer 3 Report

Dear Sirs, 
In my opinion Your manuscriot is very interesting, You were able to gather interesting data to support You review. Cases are adequately decribed but I suggest adding in table 1 at least 3-5 more examples of the strain/case in each group(bacteria, yeast and algae) of MEM organisms used /or at least laboratory tested in lignocellulosic materials valorisation. I've also able to spot single lack of the letter in teh word "engineered" - line 259. 
I think that it requires minor changes in the text and after that it should be acepted for publication. Please see the attached file with my comments added.

Author Response

Dear Sirs,

In my opinion Your manuscriot is very interesting, You were able to gather interesting data to support You review. Cases are adequately decribed but I suggest adding in table 1 at least 3-5 more examples of the strain/case in each group(bacteria, yeast and algae) of MEM organisms used /or at least laboratory tested in lignocellulosic materials valorisation. I've also able to spot single lack of the letter in teh word "engineered" - line 259.

Response: This observation coincided with one of Reviewer 2. We implemented both together as described above.

I think that it requires minor changes in the text and after that it should be acepted for publication. Please see the attached file with my comments added.

Response: We downloaded the file and corrected all observations made.